# The Sustainability of Family Ownership on the Choice of Foreign Market Entry Mode: Empirical Evidence from Listed Family Firms in China

Qingnian Wang [1,2,*] , Yunpei Wang [3], Xiaoping Li [4] and Lan Tang [5]

1 School of Economics and Finance, South China University of Technology, Guangzhou 510006, China
2 School of International Education, South China University of Technology, Guangzhou 510006, China
3 School of Economics and Management, South China Normal University, Guangzhou 510631, China; 20210733012@m.scnu.edu.cn
4 Dongguan Power Supply Bureau, Guangdong Power Grid Company Ltd., Guangzhou 510663, China; lixiaoping0625@163.com
5 School of Journalism and Communication, South China University of Technology, Guangzhou 510006, China
* Correspondence: qnwang@scut.edu.cn

**Abstract:** Family firms make up the majority of private firms in China and play an important role in China's national economy. With the deepening development of globalization and the implementation of the "going global" strategy, the overseas investment of family firms in China is increasing day by day. In the process of overseas investment, family firms often face the choice of equity entry mode. And, family strategic decisions may be influenced by family characteristics, in which family ownership is the key. Therefore, this paper discusses how family ownership affects the choice of equity entry mode in the overseas market of family firms. Based on social emotional wealth theory, this paper tries to discuss the relationship between family ownership and equity entry mode of Family firms, bring in external environment and internal governance factors of family firms, and put forward a research hypothesis. In order to verify the hypothesis, this paper takes 623 A-share listed family firms in the Shanghai and Shenzhen stock markets of China from 2010 to 2018 as research samples and tests the data through binomial logistic regression. The findings are as follows: (1) There is a positive correlation between family ownership and the entry mode of family firms in overseas markets. (2) Both the investment uncertainty of a host country and the shareholding ratio of institutional investors negatively moderate the positive correlation between family ownership and the shareholding entry mode of family firms in overseas markets. (3) The quality of home and regional institutions positively moderates the relationship between family ownership and family firms' equity entry mode in overseas markets. The conclusions expand the empirical research on the relationship between the heterogeneity of Chinese family firms, the strategy of equity entry mode, and their sustainability.

**Keywords:** family firms; family ownership; social emotional wealth theory; equity entry mode; sustainability

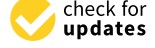



## 1. Introduction

As a very common form of enterprise organization in the social economy, family enterprises are numerous and play an increasingly important role in the global economy. In China, family enterprises are an important part and backbone of private enterprises and play an indispensable role in the vigorous development of China's national economy. In recent years, with the deepening of China's "going out" strategy, more and more family enterprises in China now participate in overseas investment activities.

For multinational family enterprises, it is very important to choose the mode of equity entry in overseas market. Multinational family enterprises need to invest resources when

entering overseas markets, which will involve costs and time. Therefore, once the entry mode is established, it is difficult to change or correct it [1–3]. In addition, the choice of entry mode has a profound impact on the formulation of other operational strategies and enterprise performance after entering the market. Family characteristics may influence the behavior and strategy of the enterprise to a certain extent. Family ownership is a very important feature of family business, a facet of heterogeneity of family business, and a symbolic expression of social emotional wealth [4]. However, there is still a lot of research space on the role of the characteristics and heterogeneity of family firms in their entry mode selection. Most foreign scholars take family enterprises in developed economies as samples and pay less attention to the mode of equity entry in overseas markets of family enterprises in emerging economies [5].

Family business is very common in China. Because of the influence of traditional family culture, Chinese family business has very obvious and distinctive family characteristics. At the same time, China is still in the period of economic transition, and different regions have different degrees of marketization, so the quality of the system is uneven. It is also worth discussing how the regional system of the home country affects family enterprises' choice of overseas market equity entry mode. Secondly, in the overseas market, family enterprises will also face the investment uncertainty brought by the host country's politics, economy, culture, and other aspects. Thirdly, in the case of high investment uncertainty in the host country, whether family firms will change their equity entry mode strategy in overseas markets and what the influencing mechanism is remains to be studied. From the perspective of corporate governance, the existence of institutional investors promotes good corporate governance and greatly influences the strategic choices of companies. Fourthly, faced with the choice of equity entry mode in overseas markets, they may tend to make strategic decisions that reflect the maximization of shareholders' interests, and their goals may be inconsistent with those of family owners. Whether this affects the choice of equity entry mode in overseas markets of family enterprises will be studied in this work. In addition, the intergenerational inheritance and family time in the characteristics of family enterprises may also have an impact on corporate strategic choice. Therefore, further analysis of the intergenerational inheritance and family time can promote the selection of equity entry mode in the overseas market of family enterprises.

## 2. Literature Review, Theoretical Analysis, and Hypotheses

### 2.1. Research on Family Business and Ownership

The academic research on Family Business started earlier. Lansberg (1988) raised the question "What is a family business" in Family Business Review, a professional academic journal that discusses the dynamics of family business [6]. Many scholars define family business based on different research purposes and perspectives. Based on a review of the relevant literature, this paper summarizes several perspectives from which scholars define family business. The definition of 10% ownership was adopted by Maury (2006), Chrisman (2012), and Singla (2014) [7–9]. From the perspective of family ownership and family management, Gomez-Mejia (2003) believes that two conditions are required to define a family business: one is that two or more directors have family relations; the other is that family members own or control at least 5% of the ownership of the business [10]. Chrisman (2012) also believes that the definition of family business should not only be defined from a single aspect but should consider the realistic characteristics of family business in a more comprehensive way; otherwise, the goal of clearly and accurately defining family business cannot be achieved [11]. His definition of family business combines the perspectives of family ownership, family management, and intergenerational inheritance. In his empirical research, he defines family business as an enterprise in which family members participate in the ownership and management of the enterprise and hope to control it across generations. Sciascia (2012) proved, using an empirical method, that there is an inverted U-shaped relationship between family ownership and the degree of internationalization of family

firms [12], while Santulli (2019) came to the conclusion that there is a U-shaped relationship between the two [13].

Masset et al. (2019) found that both family and non-family blockholders displayed their higher use for assets in the lodging industry [14]. Some scholars (Giménez et al., 2020; Guo 2022) studied the problems of family firm succession using different theories, such as Unified and Synthesized theory, Corporate Social Responsibility theory, etc. [15,16]. Dong et al. (2022) analyzed 610 Chinese manufacturing family firms from 2009 to 2017. The regression analysis indicated that there was an inverted U-shaped relationship, which linked with R&D and policy [17].

### 2.2. Definition and Classification of Overseas Market Entry Mode

Sharma (2004) defines an entry mode as "a structured agreement in which a company implements its product market strategy in international markets by itself (export, sole proprietorship) or in partnership with others (contract mode, joint venture)" [18]. Mart et al. (2021) analyzed the performance of firms and the relationship between cooperative R&D and political ties in the Gulf Cooperation Council (GCC) countries [19]. Mondal et al. (2022) analyzed family ownership in the multi-national context in India and suggested local subsidiaries for the multi-national field [20]. Levesque et al. (2022) believed that powerful planning was crucial to family firms [21].

### 2.3. The Theoretical Basis of Equity Entry Mode and Influencing Factors in Overseas Market

### 2.3.1. Transaction Cost Theory

Transaction cost theory is often used in the research of entry mode. The basic principle of this theory is that multinational enterprises will choose the entry mode that minimizes transaction costs and maximizes efficiency and revenue. This theory was proposed by Williamson (1985), who believed that transaction is the basic unit of organizational analysis and "the cost of running the economic system", and the pursuit of the lowest transaction cost is the fundamental principle of this theory [22]. In Williamson's theoretical framework, the factors that determine the level of transaction cost include asset specificity, uncertainty (internal behavior and external market uncertainty), and transaction frequency.

### 2.3.2. Institutional Theory

North (1990), a representative of the institutional theory, believes that the strategic actions of enterprises are highly dependent on and rooted in the system. The environment is fundamentally influenced by both formal and informal rules [23]. Specifically, formal rules (such as laws, regulations, or policies) and informal rules (such as norms, ethics, beliefs, and culture) can shape a firm's resources and influence the formation of a firm's competitive advantage, thus significantly influencing firm behavior. Li et al. (2022) used a novel dataset to follow the evolution of family ownership, firm value, and firm policies for up to 25 years post-initial public offering (IPO). Firm value, measured by Tobin, is fundamentally influenced by both formal and informal rules [23], including formal rules (such as laws, regulations, or policies) and informal rules (such as from activity to internal cash flow [24]). Ghalke et al. (2023) analyzed the relationship between ownership and performance and found a positive link [25].

### 2.3.3. Social Emotional Wealth Theory

Social Emotional Wealth Theory, also known as SEW theory, was proposed by Gomez-Mejia (2007) [26]. The theory holds that family businesses attach great importance to the accumulation of social emotional wealth, rather than merely pursuing the maximization of economic interests. Social emotional wealth refers to the emotional endowment that family members obtain from the family business, which is a kind of non-economic utility to meet the emotional needs of the family. According to Gomez-Mejia, social emotional wealth includes maintaining family control and influence over the business, appointing trusted family members to important positions, establishing and perpetuating family culture and

values in the business, and achieving the goal of transferring power to future generations of the family. The theory of social emotional wealth has opened up new ideas for the strategic management and decision making of family enterprises and has been applied to many topics of family enterprise research, such as the social responsibility performance of family enterprises, acquisition behavior, R&D investment, internationalization, etc., becoming an important theoretical basis for the strategic decision making of family enterprises.

The social emotional wealth theory developed from the behavioral agency theory, which was proposed by Gomez-Mejia et al. [27]. According to this theory, a company's strategic decisions depend on key decision makers, who will make corresponding decisions in order to preserve and maintain their interests or endowments in the company [28]. In the context of a family business, family members are the key decision makers in the family business because they have greater decision-making power. They mainly consider the degree of goal realization of social emotional wealth to make evaluation decisions. When this emotional endowment cannot be satisfied, family members tend to make decisions that are not driven by economic factors, that is, they may be willing to accept financial loss risk to prevent the loss of social emotional wealth [29,30].

After Gomez-Mejia proposed the theory of social emotional wealth, Berrone et al. (2012) [4] proposed the concept of the five-dimensional structure of social emotional wealth through further research, thus deepening the academic community's understanding of the theory of social emotional wealth. The first dimension is family control and influence. This dimension is an integral part of the emotional wealth of society and is highly desired by family members. The second dimension is family members' recognition of the company. Family members place a high value on business credibility because the family business is seen by internal and external stakeholders as an extension of the family in another form, a projection of the family's core values. The third dimension refers to the social relations of the family business. Reciprocal relationships within a family business are not limited to family members but may extend to a wider group. The fourth dimension involves family members' emotional connection to the business. This sentiment permeates the organization and influences the decision-making process of the family business. The fifth dimension is the intention to pass on the business to future generations. Family members view the family business as a long-term family investment and tend to pass it on to future generations.

Umas et al. (2023) examined firm performance based on the behavioral theory with a sample of 209 family firms, and they found that the level of succession planning was higher when family firm performance was further below historical aspirations [31]. Saeed et al. (2023) found that the fifth dimension was the intention to pass on the business to ISO 14,001 certification, and this dimension was in tiny firms [32].

### 2.3.4. Country-Level Factors

Institutional theory suggests that a country's institutional environment influences the choice of firm boundaries because the environment reflects the "rules of the game" for firms to participate in a particular market. A key issue for multinational enterprises to face when making resource commitment decisions in overseas markets is "how to deal with the institutional environment" [33,34]. Scholars (Setiawan et al., 2022; Pipatanantakurn et al., 2022) examined the firm performance from the data of their own countries based on a country-level sample [35,36].

### 2.3.5. Industry-Level Factors

In a study of the intensity of competition in the host country industry and the mode of equity entry in overseas markets, Bell (1996) pointed out that in highly competitive host country industries, multinational enterprises are more willing to focus on developing their specific competitiveness through sole-ownership entry mode [37]. Scholars (Cisneros et al., 2022; Hsu et al., 2023) examined the external social capital, social emotional wealth from the industry level and showed that they were positive [38,39].

### 2.3.6. Firm-Level Factors

From the micro level, scholars mainly study the equity entry mode of enterprises in overseas markets from the aspects of enterprise nature, proprietary technology, enterprise scale, financial performance, international experience, and so on [40,41]. Zhou (2017) argues that the investment behavior of state-owned enterprises follows the "state logic", and their decisions reflect the political goals set by the government to varying degrees [42]. Scholars (Rosecks et al., 2022; D'Este et al., 2022; Reddy et al., 2023) examined social capital and firm performance from the firm level and found that family conflict and "risk-taking" harmed performance [43–45].

### 2.4. Research on Family Business and Equity Entry Mode in Overseas Market

Abdellatif (2010) took 759 Japanese companies as research samples, which were divided into family enterprises and non-family enterprises. Research shows that family firms prefer joint ventures less than non-family firms. This is because the joint venture model, as opposed to sole proprietorship, requires the company to consider the preferences of local partners in any major decision, which is contrary to the desire of family businesses to maintain family control and independent decision making [46]. Ulrich et al. (2014) investigated the entry patterns of Danish family and non-family firms into BRIC markets using a sample of 177 Danish SMEs. The research finds that, compared with non-family firms, family firms choose the high-commitment entry mode, namely the sole proprietorship mode, in order to establish a more lasting relationship in the host country and maintain the family's control and influence in the subsidiary company and other social emotional wealth. Non-family firms, however, tend to prefer joint venture models with greater flexibility and less control when entering BRIC markets [47,48]. Sestu et al. (2018) combined the social–emotional wealth theory and transaction cost theory, and they found that the decision making of equity entry mode of Italian family and non-family firms would be affected by the family or non-family nature of local firms [49]. Other scholars (Sena et al., 2022; Maggi et al., 2023) examined institutional variables and multinational firms and were concerned about their equity entry mode [50,51].

In general, this paper analyzed different aspects and methods of application to study the literature, and the main list of the literature review is summarized in Table 1.

**Table 1.** The main list of literature review.

| Year | Fields and Main Points | Theoretical | Empirical | Researchers' Name |
|------|------------------------|-------------|-----------|-------------------|
| | *Theme 1: Research on Family Business and Ownership* | | | |
| 1988 | Concept of family business | √ | | Lansberg [6] |
| 2003 | Family relations | √ | | Gomez-Mejia [10] |
| 2006 | Family Ownership | √ | | Maury [7] |
| 2012 | Family Ownership | √ | | Chrisman [11] |
| 2012 | Inverted U-shaped relationship | √ | √ | Sciascia [12] |
| 2014 | Family Ownership | √ | | Singla [9] |
| 2019 | Relationship between blockholder ownership, asset levels, and corporate performance | | √ | Masset et al. [14] |
| 2020 | Theory of family firm succession | √ | | Giménez et al. [15] |
| 2022 | Firm's initiated successions and succession | | √ | Guo [16] |
| 2022 | R&D and policy | | √ | Dong et al. [17] |
| | *Theme 2: Definition and Classification of Overseas Market Entry Mode* | | | |
| 2004 | Defines an entry mode | √ | | Sharma [18] |
| 2021 | Analyzed the effect of blockholders and interaction | √ | √ | Martínez-Garcia et al. [19] |
| 2022 | Dimensions of family firm multi-nationality | √ | √ | Mondal et al. [20] |

**Table 1.** *Cont.*

| Year | Fields and Main Points | Theoretical | Empirical | Researchers' Name |
|---|---|:---:|:---:|---|
| 2022 | Succession roadmap | √ | √ | Levesque et al. [21] |
| | *Theme 3: Theoretical Basis of Equity Entry Mode in Overseas Market* | | | |
| 1985 | Minimizes transaction costs and maximizes efficiency and revenue | √ | | Williamson [22] |
| 1990 | The institutional theory, The environment is influenced by both formal and informal rules | √ | | North [23] |
| 2022 | The evolution of family ownership | | √ | Li et al. [24] |
| 2023 | The endogeneity of the ownership-performance relationship | | √ | Ghalke et al. [25] |
| 2007 | Social Emotional Wealth Theory | √ | | Gomez-Mejia [26] |
| 2012 | The concept of the five-dimensional structure of social emotional wealth | √ | | Berrone et al. [4] |
| 2023 | Succession planning activities in family firms | | √ | Umas et al. [31] |
| 2023 | Preserve family legitimacy and socio-emotional wealth | √ | √ | Saeed et al. [32] |
| | *Theme 4: Different levels' Influencing Factor in Overseas Market* | | | |
| 2022 | Effect of corporate social responsibility (CSR) on firm performance | | √ | Setiawan et al. [35] |
| 2022 | Successors and predecessors | | √ | Pipatanantakurn et al. [36] |
| 1996 | Multinational enterprises chose sole ownership entry mode | √ | | Bell [37] |
| 2022 | The transfer of external social capital from a predecessor | | √ | Cisneros et al. [38] |
| 2023 | Family control and succession | | √ | Hsu et al. [39] |
| 2017 | Investment behavior of state-owned enterprises | | √ | Zhou [42] |
| 2022 | Role of family social capital, conflict and, firm performance | | √ | Rosecká et al. [43] |
| 2023 | Family managers' influence on firms' risk choices | | √ | D'Este et al. [44] |
| 2023 | Quadratic inverse-U relationship | √ | √ | Reddy et al. [45] |
| | *Theme 5: Research on Family Business and Equity Entry Mode in Overseas Market* | | | |
| 2010 | Considering the preferences of local partners | | √ | Abdellatif [46] |
| 2014 | The entry patterns of Danish family and non-family firms | | √ | Ulrich et al. [47] |
| 2018 | Combined the social–emotional wealth theory and transaction cost theory | √ | √ | Sestu et al. [49] |
| 2022 | Impact of institutional variables on multinational enterprises' choice | √ | √ | Sena et al. [50] |
| 2023 | Literature review | | √ | Maggi et al. [51] |

*2.5. Research Hypothesis*

2.5.1. Hypothesis 1: Family Ownership Is Positively Correlated with the Equity Entry Mode of Family Enterprises in the Overseas Market

From the perspective of family control and influence, the family is more likely to pursue the mode of complete family control in its strategic decision making and choose the sole proprietorship mode.

From the perspective of altruistic opportunities to family members, the appointment of family members can coordinate interests and reduce information asymmetry, thus reducing governance costs [52]. In addition, by appointing family members to participate in the management of overseas subsidiaries, permanent human capital, social capital, and viability capital are created [53] to positively influence the performance of overseas subsidiaries. Therefore, family enterprises with high family ownership tend to choose the mode of entering the overseas market with sole proprietorship in order to have the independent right of personnel appointment of overseas subsidiaries.

From the perspective of the continuation of family values in enterprises, family enterprises with high family ownership are more inclined to adopt the mode of sole proprietorship to enter the overseas market in order to unify the vision and goals of the family with the strategy they follow and maintain their influence on overseas subsidiaries. In terms of improving family reputation, the higher the family ownership, the stronger the

motivation to pursue good reputation, and the more willing to choose the entry mode of sole proprietorship in overseas investment.

Therefore, this paper proposes hypothesis H1: Family ownership is positively correlated with the equity entry mode of family enterprises in the overseas market.

### 2.5.2. Hypothesis 2: The Moderating Role of Investment Uncertainty in Host Country

The investment environment of a host country affects the choice of entry mode, survival, and development of family enterprises in overseas markets and requires family enterprises to adapt to the change in environment through strategic adjustment.

When the host country investment increases uncertainty, in order to reduce the investment of the transaction costs and adverse effects of uncertainty, multinational companies will reduce the resource commitment in the host country to avoid the higher cost and risk, so the family firm pursuit of family control tendency might be to reduce social emotions, such as wealth, and will choose a more flexible mode than joint venture into the market.

Therefore, we propose hypothesis H2: investment uncertainty in a host country has a negative moderating effect on the relationship between family ownership and equity entry mode of family enterprises in overseas market.

### 2.5.3. Hypothesis 3: The Moderating Effect of Regional Institution Quality in Home Country

Due to the gradual characteristics of China's economic and institutional reform, there are considerable institutional differences among different regions in China. In the context of such a large economic entity as China, the quality of the regional institution of the home country mainly affects the equity entry mode of family enterprises in the overseas market through three aspects: product market, factor market, and market intermediary organization and legal system.

First of all, a well-functioning product market can help family enterprises improve production and operation efficiency and enhance their confidence and ability for overseas investment, so they tend to choose the mode of entering the overseas market of sole proprietorship. In addition, developed factor markets also help family businesses raise funds to invest overseas through the financial sector. Finally, a good market intermediary organization and perfect legal environment jointly guarantee the survival and development of family enterprises in the domestic market, so that family enterprises can save the use of capital and other factors' input, meaning that they have sufficient capital and ability to increase resource commitment when entering the overseas market.

To sum up, a higher quality of regional institutions in the home country can motivate local family firms to improve efficiency and competitiveness, enhance their confidence in maintaining social emotional wealth, and strengthen their tendency to choose the sole proprietorship mode.

Therefore, hypothesis H3 is proposed: the quality of home country regional institutions has a positive moderating effect on the relationship between family ownership and ownership entry mode of family firms in overseas markets.

### 2.5.4. Hypothesis 4: The Regulating Effect of the Shareholding Ratio of Institutional Investors

As equity owners of a company, institutional investors may influence the strategy of a family business. When facing the mode of equity entry in overseas markets, unlike family owners, institutional investors do not make decisions based on social emotional wealth goals such as family control. Instead, they make decisions based on corporate resources and capabilities and balance between income, investment cost, and uncertainty.

The size of the voice of institutional investors depends on their shareholding ratio. When the shareholding ratio of institutional investors is larger, their speaking power will also become larger, and their ability to influence the strategic decisions of the family business will become stronger.

Therefore, hypothesis H4 is proposed: the shareholding ratio of institutional investors has a negative moderating effect on the relationship between family ownership and equity entry mode of enterprises in overseas market.

## 3. Methods and Design

This section selects all a-share listed family companies in China's Shenzhen and Shanghai stock markets from 2010 to 2018 as the research samples, takes the overseas market equity entry mode as the dependent variable and family ownership as the independent variable, and selects five indicators from the political, economic, and cultural aspects of the host country to measure the investment uncertainty.

### 3.1. Variable Definitions

Specific variable definitions are shown in Table 2.

**Table 2.** Variable definitions and data sources.

| Variable Type | Variable Symbol | Variable Name | Variable Description | Data Source |
|---|---|---|---|---|
| Dependent Variable | Mode | Overseas Market Equity Entry Mode | If an enterprise owns 95% or more of the equity of an overseas subsidiary, it is a sole proprietorship entry mode, and the value is 1; otherwise, it is 0 | CSMAR database |
| Independent Variable | FO | Family Ownership | Actual control Proportion of ultimate family ownership in the business | CSMAR database |
| Regulated Variable | Unc | Host Country Investment Uncertainty | Factor analysis was used and formula (4–5) was used to calculate | the World Bank, Hofstede website, CEPII database |
| | Iq | Home Country Regional Institutional Quality | Marketization Index | Marketization Index Developed by Wang Xiaolu and Others |
| | InsInvest | Proportion of shares held by InsInvest institutional investors | Proportion of shares held by all institutional investors to total shares | CSMAR database |
| Control Variable | Size | Parent Company Size | The natural logarithm of the total assets of the parent company | CSMAR database |
| | Age | Parent Company Age | observation year minus the natural logarithm of the year of establishment of the enterprise | CSMAR database |
| | Lev | Parent Company Financial Leverage | Asset-Liability Ratio | CSMAR database |
| | ROA | Parent Company Financial Performance | Return on Assets | CSMAR database |
| | Resource | Host Country Resources | Host Country Share of exports of fuels, metals and ore products in total exports | the World Bank |
| | Market | Host Country Market Size | Host Country Gross National Product | the World Bank |
| | Open | Openness of the Host Country's Market | Proportion of the Host Country's Trade Volume to GDP | the World Bank |
| | Ind | Industry | manufacturing is 1, non-manufacturing is 0 | |
| | Year | Year | Annual dummy variable | |

*3.2. Empirical Model*

Based on the relationship between family ownership and equity entry mode of family enterprises in overseas markets and the role of three moderating factors, a regression model is constructed. This paper studies the modes of equity entry of Chinese family enterprises in overseas market, including sole proprietorship and joint venture. Sole proprietorship is represented by 1, and joint venture is represented by 0. This paper adopts binomial logistic regression model to conduct empirical research. The model settings of this paper are as follows:

$$P(Y) = \frac{1}{1 + e^{-(\beta_0 + \sum_{k=1}^{n} \beta_k X_k)}} \text{ (1-1)}$$

where Y is the dependent variable overseas market equity entry mode, $\beta_0$ is the intercept, $\beta_1, \beta_2 \ldots \beta_n$ is the regression coefficient, and $X_1, X_2, \ldots X_n$ is the explanatory variable. In the empirical regression process, if the regression coefficient is positive, it indicates that this explanatory variable increases the probability of family enterprises choosing the sole proprietorship mode to enter the overseas market. Otherwise, the probability is reduced.

*3.3. Data Acquisition*

After comprehensively considering the availability and matching degree of data, this paper selects the family enterprises listed in Shanghai and Shenzhen during 2010–2018 as the research object. The data calculation and analysis process was mainly completed using stata14.0. After matching and screening, a total of 7331 observation data of 623 listed family enterprises were collected.

The time window of sample selection in this paper covers a total of nine years from 2010 to 2018, and a total of 7331 observation data of 623 listed family enterprises in China were obtained. From the year distribution of the samples, it can be seen that the number of listed family enterprises' overseas investment samples has increased year by year, from 95 samples in 2010 to 2001 samples in 2018. It can be seen that in recent years, more and more family enterprises choose to go abroad and expand their overseas territory. Especially in the four years from 2015 to 2018, the sample size of the four years accounted for 11.50%, 16.04%, 23.27%, and 27.30% of the total sample, respectively, accounting for 78.11% of the total sample, indicating that the momentum of overseas investment of listed family enterprises in these four years is strong.

## 4. Results

*4.1. Empirical Results*

4.1.1. Multicollinearity Test

In order to exclude the influence of multicollinearity, this paper uses variance inflation factor (VIF) to perform multicollinearity diagnostic estimates for the main variables in the model (as shown in Table 3). The results show that the mean value of variance inflation factor is 1.49 and the maximum value is 2.07, both of which are less than 10. This shows the data are suitable for further study.

**Table 3.** Results of variance inflation factor.

| Variable | VIF |
|:---:|:---:|
| Market | 2.07 |
| Open | 2.05 |
| Lev | 1.79 |
| Size | 1.61 |
| ROA | 1.26 |
| Resour | 1.1 |
| FO | 1.04 |
| Age | 1.02 |
| Mean VIF | 1.49 |

### 4.1.2. Main Effect Regression

Binomial logistic regression analysis is also used to test the influence of family ownership on the equity entry mode of family firms in overseas markets and the moderating effect of three factors (investment uncertainty of host country, regional institutional quality of home country, and shareholding ratio of institutional investors) (Table 4).

**Table 4.** Regression results of main effects.

| Variable | Dependent Variable: Mode (Sole Proprietorship = 1) | |
| --- | --- | --- |
| | Model (1) | Model (2) |
| FO | | 0.8259 *** |
| | | (4.1505) |
| Size | 0.1739 *** | 0.1908 *** |
| | (5.1666) | (5.6670) |
| Age | 0.0053 | 0.0075 |
| | (0.8364) | (1.1741) |
| Lev | −0.6361 *** | −0.6395 *** |
| | (−3.0935) | (−3.0621) |
| ROA | −0.5112 * | −0.5582 * |
| | (−1.7423) | (−1.8729) |
| Resour | −0.0049 ** | −0.0048 ** |
| | (−2.4171) | (−2.3638) |
| Market | 0.1846 *** | 0.1871 *** |
| | (7.3101) | (7.4581) |
| Open | 0.0027 *** | 0.0027 *** |
| | (5.1657) | (5.2095) |
| Constant | −7.5348 *** | −8.3135 *** |
| | (−7.0189) | (−7.7441) |
| Vintage effect | YES | YES |
| Industry effect | YES | YES |
| Pseudo R2 | 0.0194 | 0.0218 |
| Log likelihood | −3670.4292 | −3661.6352 |
| Prob > chi2 | 0.0000 | 0.0000 |
| Chi2 | 156.57 | 179.95 |
| N | 7331 | 7331 |

*** means $p < 0.01$, ** means $p < 0.05$, * means $p < 0.1$.

### 4.1.3. Moderating Effect Regression

In Table 5, Model (3) introduces host country investment uncertainty (Unc) and its interaction with family ownership (FO) on the basis of Model (2) in Table 5 to verify the moderating effect of host country investment uncertainty. The regression results show that the interaction coefficient of family ownership and host country investment uncertainty is −2.5309, which is significantly correlated at the level of 10%. It can be seen that the investment uncertainty of the host country has a negative moderating effect on the relationship between family ownership and the equity entry mode of family firms in overseas markets, and hypothesis 2 of this paper is verified. This indicates that although family firms attach importance to social emotional wealth and prefer the entry mode with high control, they will carefully consider the market entry strategy when entering a host country with high investment uncertainty. According to the transaction cost theory, when investment uncertainty is high, the cost of acquiring information is higher, and the barrier to acquiring social networking is greater. The joint venture model can reduce the negative

impact caused by investment uncertainty. On the one hand, because the joint venture model involves two or more enterprises, the multinational enterprise can share the cost and risk with the local partner, limiting the risk to the investment share. On the other hand, by working with local partners, family businesses can have a better understanding of local market patterns. In addition, to some extent, family firms can gain competitive advantages from cooperative enterprises' organizational learning and knowledge management to cope with investment uncertainties in host countries. Therefore, in the face of high uncertainty in the host country environment, family companies are willing to break the dominant logic of family control and give up part of the control of overseas subsidiaries to enter overseas markets.

**Table 5.** Regression results of moderating effects.

| Variable | Dependent Variable: Mode (Sole Proprietorship = 1) | | |
|---|---|---|---|
| | **Model (3)** | **Model (4)** | **Model (5)** |
| FO | 0.9105 *** | 1.6749 *** | 1.3979 *** |
| | (4.3391) | (2.7227) | (3.9888) |
| Unc | 0.7104 | | |
| | (1.0657) | | |
| Iq | | −0.0659 | |
| | | (−1.4760) | |
| InsInvest | | | 1.2169 *** |
| | | | (3.5287) |
| FO × Unc | −2.5309 * | | |
| | (−1.7201) | | |
| FO × Iq | | 0.3663 *** | |
| | | (3.4072) | |
| FO × InsInvest | | | −1.8081 ** |
| | | | (−2.4322) |
| Size | 0.1908 *** | 0.1923 *** | 0.1573 *** |
| | (5.6489) | (5.7369) | (4.4242) |
| Age | 0.0071 | 0.0087 | 0.0027 |
| | (1.1052) | (1.3612) | (0.4184) |
| Lev | −0.6422 *** | −0.6708 *** | −0.6480 *** |
| | (−3.0688) | (−3.2094) | (−3.1074) |
| ROA | −0.5683 * | −0.5549 * | −0.7316 ** |
| | (−1.9015) | (−1.8818) | (−2.2836) |
| Resour | −0.0058 *** | −0.0042 ** | −0.0045 ** |
| | (−2.7489) | (−2.0874) | (−2.2268) |
| Market | 0.1708 *** | 0.1928 *** | 0.1794 *** |
| | (5.6069) | (7.6411) | (7.0977) |
| Open | 0.0026 *** | 0.0027 *** | 0.0026 *** |
| | (4.8353) | (5.1460) | (5.1046) |
| Constant | −7.8456 *** | −7.7226 *** | −7.7260 *** |
| | (−6.5524) | (−6.7866) | (−6.9498) |
| Vintage effect | YES | YES | YES |
| Industry effect | YES | YES | YES |
| Pseudo R2 | 0.0223 | 0.0252 | 0.0242 |
| Log likelihood | −3659.753 | −3648.9103 | −3652.5364 |
| Prob > chi2 | 0.0000 | 0.0000 | 0.0000 |
| Chi2 | 183.47 | 195.14 | 205.32 |
| N | 7331 | 7331 | 7331 |

*** means $p < 0.01$, ** means $p < 0.05$, * means $p < 0.1$.

The regression results show that all four hypotheses proposed in this paper are valid, and they all pass the robustness test.

## 4.2. Robustness Test

In this paper, the parent company holding 90% of the shares of overseas subsidiaries is set as the sole proprietorship entry mode, with the value being 1; otherwise, it is the joint venture mode, with the value being 0. The model is re-regression, and the robustness test results are in Table 6. As can be seen from the table, the regression coefficient between family ownership and family firm's equity entry mode in overseas market is significantly positive. Hypothesis 1 passes the robustness test; the adjustment variables of host country investment uncertainty, home country regional institutional quality, and institutional investor shareholding ratio are intersected with family ownership, respectively. The symbols and significance of the terms are basically consistent with the previous ones, indicating that hypotheses 2, 3, and 4 all pass the robustness test.

**Table 6.** Robustness test results for changing dependent variable criteria.

| | Dependent Variable: Mode (Sole Proprietorship = 1) | | | | |
|---|---|---|---|---|---|
| Variable | Model (1) | Model (2) | Model (3) | Model (4) | Model (5) |
| FO | | 1.0379 *** | 1.1383 *** | 1.8541 * | 2.2948 *** |
| | | (5.0254) | (5.2349) | (1.8380) | (6.2563) |
| Unc | | | 0.9431 | | |
| | | | (1.3502) | | |
| Iq | | | | −0.0066 | |
| | | | | (−0.1428) | |
| InsInvest | | | | | 1.6032 *** |
| | | | | | (4.5467) |
| FO × Unc | | | −2.8761 * | | |
| | | | (−1.8686) | | |
| FO × Iq | | | | 0.2934 *** | |
| | | | | (2.6445) | |
| FO × InsInvest | | | | | −3.3791 *** |
| | | | | | (−4.4738) |
| Size | 0.1761 *** | 0.1962 *** | 0.1960 *** | 0.1959 *** | 0.1791 *** |
| | (4.9648) | (5.5150) | (5.4878) | (5.5476) | (4.8189) |
| Age | 0.0171 *** | 0.0204 *** | 0.0199 *** | 0.0227 *** | 0.0166 ** |
| | (2.5782) | (3.0206) | (2.9375) | (3.3790) | (2.4132) |
| Lev | −0.4062 * | −0.4010 * | −0.4064 * | −0.4444 ** | −0.4540 ** |
| | (−1.9013) | (−1.8413) | (−1.8621) | (−2.0480) | (−2.0716) |
| ROA | −0.5608 | −0.6163 * | −0.6351 * | −0.5918 * | −0.7494 * |
| | (−1.5805) | (−1.6787) | (−1.7190) | (−1.6684) | (−1.9313) |
| Resour | −0.0042 ** | −0.0040 * | −0.0048 ** | −0.0032 | −0.0035 * |
| | (−1.9907) | (−1.9067) | (−2.1924) | (−1.5389) | (−1.6731) |
| Market | 0.1643 *** | 0.1675 *** | 0.1574 *** | 0.1727 *** | 0.1623 *** |
| | (6.2858) | (6.4548) | (5.0264) | (6.6219) | (6.2092) |
| Open | 0.0023 *** | 0.0023 *** | 0.0022 *** | 0.0023 *** | 0.0023 *** |
| | (4.3068) | (4.3544) | (4.1136) | (4.2390) | (4.3057) |
| Constant | −7.1272 *** | −8.0895 *** | −7.8153 *** | −7.9231 *** | −8.1134 *** |
| | (−6.3554) | (−7.1998) | (−6.2707) | (−6.6870) | (−7.0227) |
| Vintage effect | YES | YES | YES | YES | YES |
| Industry effect | YES | YES | YES | YES | YES |
| Pseudo R2 | 0.0185 | 0.0221 | 0.0226 | 0.0269 | 0.0250 |
| Log likelihood | −3468.7788 | −3455.9979 | −3454.1564 | −3438.7787 | −3445.6084 |
| Prob > chi2 | 0.0000 | 0.0000 | 0.0000 | 0.0000 | 0.0000 |
| Chi2 | 142.64 | 175.06 | 177.25 | 198.63 | 198.03 |
| N | 7331 | 7331 | 7331 | 7331 | 7331 |

*** means $p < 0.01$, ** means $p < 0.05$, * means $p < 0.1$.

## 4.3. Research Analysis and Conclusions

There is a positive correlation between family ownership and the equity entry mode of family enterprises in overseas market; that is, the higher the proportion of family

ownership, the more inclined the family enterprises are to choose the sole proprietorship entry mode. When family businesses make big strategic decisions, they are motivated by the desire to preserve or enhance social emotional wealth. Therefore, sole proprietorship is preferred to enter the mode to prevent external forces from diluting the family's control and influencing the decision making of the family business. When there is a family member in the management of an overseas subsidiary, the family enterprise will have a stronger ability to monitor the overseas subsidiary, so as to reduce the agency cost related to investment in overseas enterprises and ensure the realization of family interests. As family ownership increases, they prefer the foreign market entry mode of sole proprietorship to better protect their goals of social emotional wealth.

Investment uncertainty in a host country has a negative moderating effect on the relationship between family ownership and equity entry mode of family enterprises in overseas markets; that is, the higher the investment uncertainty of a host country, the weaker the positive correlation between family ownership and equity entry mode of family enterprises in overseas market.

The quality of regional institutions in the home country positively moderates the relationship between family ownership and equity entry mode of family enterprises in overseas markets; that is, the better the quality of regional institutions in the home country, the stronger the positive correlation between family ownership and equity entry mode of family enterprises in overseas market. If a family enterprise is located in a region with a high-quality regional system, with an improvement in family ownership, it will have a stronger preference for the entry mode of sole proprietorship.

The shareholding ratio of institutional investors has a negative moderating effect on the relationship between family ownership and the equity entry mode of family enterprises in overseas markets; that is, the higher the shareholding ratio of institutional investors, the weaker the positive correlation between family ownership and the equity entry mode of family enterprises in overseas markets. The increase in the shareholding ratio of institutional investors will weaken the tendency of family ownership to enter the mode of sole proprietorship.

With the extension of family ownership, family members will become more and more attached to the family business. The more social emotional wealth they accumulate, the more they hope to maintain control and influence over the business and tend to enter the overseas market in the form of sole proprietorship.

## 5. Discussion

### 5.1. Academic Implications

By studying the influence of family ownership on the equity entry mode of family enterprises in overseas markets, this paper analyzes the moderating effect of investment uncertainty of a host country, regional institutional quality of the home country, and shareholding ratio of institutional investors, as well as the influence of family characteristics, i.e., the involvement of the second generation of the family and the time of family integration, on the relationship between the two. Thus, it is critical to have a deeper understanding of the selection of equity entry mode for family enterprises in overseas markets. This has some implications for family enterprises in the period of overseas investment expansion and local governments in the period of deepening reform.

### 5.2. Managerial Implications

For family enterprises, while considering social emotional wealth, they should also strengthen their understanding of the enterprise's selection of the entry mode. In the face of increasingly fierce global competition and rapid technological development environment, multinational family enterprises may be in a disadvantageous position in terms of global competition if they only rely on their own capabilities. Therefore, where appropriate, it is necessary for multinational family businesses to complement and strengthen their knowledge and competitiveness through cooperation. Cooperation mode is not only a

form of sharing benefits and risks but also an important way for multinational enterprises to acquire knowledge and ability. Family enterprises should attach importance to the important position of institutional investors in corporate governance, take the appropriate initiative to contact institutional investors, and encourage institutional investors to actively "speak" and play a governance role.

For local governments, they should realize the importance of the institutional environment for overseas investment and development of enterprises. At present, more and more enterprises, including family businesses involved in the process of overseas investment and in the tide of the local government, should speed up marketization reform by introducing measures to increase product and factor markets, encourage the development of intermediary institutions, and promote the formulation and implementation of laws and regulations, to ensure that the enterprise can participate effectively in the market and to foster competitive advantages for enterprises to create a good "soil environment". In addition, the government should be for the concrete practice of the family enterprises to "go out" and offer effective information and policy guidance, to provide rich and useful information in overseas markets, facilitating international family firm's perception of risk in overseas markets, helping enterprises to guard against risks, having a positive impact on the formation of enterprise competitive advantage and the overseas investment process.

### 5.3. Limitations

This study has several limitations. First, in order to obtain data, only listed family businesses were selected as the research object in this study, and unlisted family businesses were not taken into account. In fact, family companies that are not listed also have overseas investment behavior and face the choice of equity entry mode in overseas markets. Future research can increase the research on the overseas market equity entry mode of unlisted family enterprises by means of questionnaire surveys and interviews. Second, the foreign investment performance of family firms remains to be studied. This paper only discusses the equity entry mode of family firms in overseas markets and does not further analyze the performance of family firms after choosing the entry mode. Tracking and analyzing the performance of family enterprises after selecting the entry mode are conducive to providing experience reference for other family enterprises to choose the equity entry mode in overseas markets. Third, this paper mainly focuses on the research of manufacturing family enterprises and does not study by industry. Therefore, the differentiated impact of family ownership on the choice of equity entry mode of family enterprises in overseas markets is limited based on different industry segments, and the effective information and targeted countermeasures and suggestions for the specific practice of family enterprises entering overseas markets under different industry backgrounds is also limited.

### 5.4. Opportunities for Further Research

Therefore, adding the latest data and expanding to unlisted enterprises in future work is critical. Furthermore, due to the differentiated impact of family ownership on the choice of equity entry mode of family enterprises in overseas markets, a further study will be focused on exploring the mechanism of equity entry mode based on different industry segments, so as to find some more effective solutions and provide more targeted countermeasures and suggestions for the specific practice of family enterprises entering overseas markets under different industry backgrounds.

**Author Contributions:** Conceptualization, Y.W., Q.W. and X.L.; methodology, Y.W., Q.W. and X.L.; validation, Y.W., Q.W., X.L. and L.T.; investigation, Y.W., Q.W., X.L. and L.T.; writing—original draft preparation, Y.W., Q.W. and X.L.; writing—review and editing, Y.W., Q.W. and L.T.; supervision, Y.W., Q.W. and L.T. All authors have read and agreed to the published version of the manuscript.

**Funding:** This study was funded by National Social Science Fund (No:21BSH097), China, the Key Project of the Center of Sino-Foreign Language Cooperation & Exchange (2021), Ministry of Education, China.

**Institutional Review Board Statement:** Not applicable.

**Informed Consent Statement:** Informed consent was obtained from all subjects involved in the study.

**Data Availability Statement:** Not applicable.

**Acknowledgments:** This study was funded by National Social Science Fund (No. 21BSH097), China, the state Project of Ministry of Science & Technology (No. DL2021163001L), China, the Key Project of the Center of Sino-Foreign Language Cooperation & Exchange(2021), Ministry of Education, China, the Industry-Academic Cooperation Collaborative Education Project of the Ministry of Education (201902036018, 202102119014), China, Guangdong Natural Sciences Fund (2020A1414010301), and the Humanities and Social Sciences Fund of South China University of Technology.

**Conflicts of Interest:** The authors declare no conflict of interest.

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
