# Peer review of "The Sustainability of Family Ownership on the Choice of Foreign Market Entry Mode: Empirical Evidence from Listed Family Firms in China"

_sustainability, doi:10.3390/su151310674_

Round 1
Reviewer 1 Report
- Advantages and benefits of the proposed approach should be given in detail. Also, the research gaps and the novelty of this study is not clear.
- The characteristics of current research should be highlighted in the comparative table of literature review from both aspects of theoretical and application.
- Generally, real data are tainted by uncertainty. The authors should discuss the proposed approach under data uncertainty.
- The authors should discuss on the limitations of the study. Also, the authors should discuss on the generalization of the results of the study. Moreover, the scientific question is not clear enough.
- The authors should compare their results and proposed approach with popular approaches in literature.
- Literature review and references should be updated according to recent studies (2021-2023).
- Please revise your conclusion part into more details. Basically, you should enhance your contributions, limitations, underscore the scientific value added of your paper, and/or the applicability of your findings/results and future study in this session. The discussion is relatively simple and insufficient.
- Some future research directions should be suggested at the end of manuscript.
The writing should be improved. Also, the paper should be re-edited to fix its errors in both punctuation and grammar.
Author Response
June 19th, 2023
Dear Professor/Reviewer,
We would like to thank you for your efficient review and the very helpful and precious suggestions. The manuscript entitled“The sustainability of Family Ownership on the Choice of Foreign Market Entry Mode: Empirical Evidence from Listed Family Firms in China”was completely revised according to your precious suggestions & comments. And the range of the paper was raised from 11 to 18 pages. The revision covered all the aspects from your suggestions and comments point by point. Please check the attached file. At the same time, the modified manuscript was re-submitted in the system.
Thank you again for your supports!
Best wishes,
Dr. Prof. Qingnian Wang

Reviewer 2 Report
This paper is a study that increased the validity of a survey based on 623 family companies listed in A shares in the Shanghai and Shenzhen stock markets in China from 2010 to 2018. However, there are some areas that need to be strengthened:
1. It needs to supplement the theoretical background more about the social-emotional health theory based on this study. Namely, it should be related to the main of this study through examples of how it was approached from the previous study and what the results of the previous studies were.
2. Page 9, in the Discussion part, it needs to check the following sentences, “I have a deeper understanding of the selection of equity entry mode for family enterprises in overseas market.” Please it needs to delete or modify that sentence. The sentence doesn't seem to relate to other contexts.
3. In the Discussion part, it is necessary to write practical implications for how to approach family enterprises from other global company in a practical implication in the future.
Because family enterprises in China are a special type compared to other countries. In other words, it is questionable how many family enterprises exist in other countries, and whether there will be many family enterprises and family management in other countries compared to the characteristic situation in China, even though the author(s) mentioned “the deepening development of globalization and the implementation of the "going global" strategy.” Therefore, it should be necessary to explain the results of this paper and how to apply them to the global market or global company.
The quality of the English Language is fine.
Author Response

(The authors gave the same response as above.)

Reviewer 3 Report
The authors should give mathematical form of the baseline Econometric Model.
There is no specification test.
The variable definitions and data sources should be in appendix.
Moderate
Author Response

(The authors gave the same response as above.)

Reviewer 4 Report
First of all, I would like to congratulate the authors for their work and effort. The paper is well written, organized and easy to understand.
Because of the existing context in China described in the paper, I understand that the object of study can be interesting and useful for this country.
Some comments and suggestions are:
- I miss some more cites that support several of the assertions made in the introduction. As well as references that serve as a basis for presupposing or justifying hypotheses 2, 3 and 4 (assertions are made without any support from the literature). Additionally, the references used, in general, are not recent, except for some from 2018 and 2019.
- More details of the sample (number, characteristics, selection...) should be given in the methodology section.
- The comments and explanations of the results obtained are too brief. With hardly any development. Perhaps it would be appropriate to join this section with the following research analysis section, and even with the Discussion section.
- The work should close with a section of conclusions after the discussion of the results. I believe that this final organization would improve the work.
Author Response

(The authors gave the same response as above.)

Round 2
Reviewer 1 Report
The characteristics of current research should be highlighted in the comparative table of literature review from both aspects of theoretical and application. In other words, a comprehensive literature review as well as research gaps should be summarized in in the comparative table.
Minor editing of English language required.
Author Response
June 24th, 2023
Dear Professor/Reviewer,
We would like to thank you again for your helpful and precious suggestions. The manuscript entitled“The sustainability of Family Ownership on the Choice of Foreign Market Entry Mode: Empirical Evidence from Listed Family Firms in China”was revised again by adding a table according to your precious suggestions & comments. The revision covered all the aspects from your suggestions and comments as the attached file.
Thank you!
Qingnian Wang
